# *Ustilago maydis* telomere protein Pot1 harbors an extra N-terminal OB fold and regulates homology-directed DNA repair factors in a dichotomous and context-dependent manner

**Syed Zahid**[1], **Sarah Aloe**[1], **Jeanette H. Sutherland**[1], **William K. Holloman**[1], **Neal F. Lue**[1,2]*

1 Department of Microbiology & Immunology, W. R. Hearst Microbiology Research Center, Weill Cornell Medicine, New York, New York, United States of America, 2 Sandra and Edward Meyer Cancer Center, Weill Cornell Medicine, New York, New York, United States of America

* nflue@med.cornell.edu

**Data Availability Statement:** All relevant data are within the manuscript and its Supporting Information files.

## Abstract

The telomere G-strand binding protein Pot1 plays multifaceted roles in telomere maintenance and protection. We examined the structure and activities of Pot1 in *Ustilago maydis*, a fungal model that recapitulates key features of mammalian telomere regulation. Compared to the well-characterized primate and fission yeast Pot1 orthologs, *Um*Pot1 harbors an extra N-terminal OB-fold domain (OB-N), which was recently shown to be present in most metazoans. *Um*Pot1 binds directly to Rad51 and regulates the latter's strand exchange activity. Deleting the OB-N domain, which is implicated in Rad51-binding, caused telomere shortening, suggesting that Pot1-Rad51 interaction facilitates telomere maintenance. Depleting Pot1 through transcriptional repression triggered growth arrest as well as rampant recombination, leading to multiple telomere aberrations. In addition, telomere repeat RNAs transcribed from both the G- and C-strand were dramatically up-regulated, and this was accompanied by elevated levels of telomere RNA-DNA hybrids. Telomere abnormalities of *pot1*-deficient cells were suppressed, and cell viability was restored by the deletion of genes encoding Rad51 or Brh2 (the *BRCA2* ortholog), indicating that homology-directed repair (HDR) proteins are key mediators of telomere aberrations and cellular toxicity. Together, these observations underscore the complex physical and functional interactions between Pot1 and DNA repair factors, leading to context-dependent and dichotomous effects of HDR proteins on telomere maintenance and protection.

## Author summary

The key function of telomeres is to stabilize chromosome ends against abnormal repair by forming a special nucleoprotein structure. One protein that plays a conserved and critical function at telomeres is Pot1, which binds to the single-stranded DNA at the very tips of chromosomes. In this study, we investigated the function of Pot1 in a yeast-like fungus

**Funding:** N.F.L. received support from the National Institute of General Medical Sciences GM107287 (https://www.nigms.nih.gov/) and from the National Science Foundation MCB-1817331 (https://www.nsf.gov/). The funders had no role in study design, data collection and analysis, decision to publish, or preparation of the manuscript.

**Competing interests:** The authors have declared that no competing interests exist.

called *Ustilago maydis*, which exhibits strong similarities to animal cells with respect to DNA repair and telomere regulation. We showed that *U. maydis* Pot1 interacts with two major DNA repair factors (Rad51 and Brh2) in a context-dependent manner. In normal cells, Pot1 binds directly to Rad51, and this interaction facilitate the maintenance of telomere DNA. However, when Pot1 is depleted and telomeres become dysfunctional, Rad51 and Brh2 become toxic to the cell by promoting abnormal recombination at telomeres. Therefore, Pot1 also negatively regulates the recombination activities of Rad51 and Brh2. These results highlight the complexity of Pot1 mechanisms and provide a compelling demonstration of the antagonistic relationship between telomere proteins and DNA repair factors. Notably, in comparison to previously characterized Pot1, *U. maydis* Pot1 has a different domain architecture, which is similar to RPA1, and which suggests an evolutionary kinship between these two protein families.

## Introduction

A key function of telomeres is to prevent chromosome ends from being recognized as double strand breaks (DSBs); erroneous recognition of telomeres as DSBs leads to the activation of DNA damage response (DDR), resulting in detrimental DNA repair (i.e., non-homologous end joining (NHEJ) or homology-directed repair (HDR)) that engenders chromosomal rearrangements [1,2]. This protective function of telomeres is largely mediated by the shelterin complex, a six-protein assembly that coats double-stranded telomere DNA repeats as well as the terminal 3' overhang (G-overhang) [1]. Among the components of shelterin are proteins that directly recognize double-stranded telomere DNA (TRF1 and TRF2), one that recognizes the G-overhang (POT1), and several associated and bridging factors (RAP1, TIN2, and TPP1). The mechanisms by which shelterin components suppress DDR signaling and DNA repair pathways and the division of labor between these components remain central questions in telomere research.

The second key function of telomeres is to promote the retention and replenishment of telomere DNA, which is subject to two types of erosion mechanisms: 1) abrupt truncation due to failure of replication forks to traverse the G/C-rich telomere repeats or DNA damage and recombination; and 2) gradual erosion due to the failure of lagging strand DNA synthesis to copy the parental DNA strand in its entirety ("end replication problem"). This DNA maintenance function of telomeres is also partly mediated by shelterin. However, in contrast to the protective function, which entails suppression of DNA repair, the DNA maintenance function often entails positive regulation of DNA repair factors or polymerases by shelterin components. For example, TPP1 serves a critical function in recruiting telomerase, a special reverse transcriptase that compensates for telomere loss by extending the G-strand [3–5]. Similarly, mammalian TRF1 is believed to recruit the BLM helicase to telomeres to help unwind G-rich secondary structures, thereby promoting replication fork progression [6]. Thus, shelterin components evidently regulate the DNA repair machinery in a context-dependent manner in order to mediate both telomere protection and telomere DNA maintenance. Understanding the molecular basis of such dichotomous regulation is a major challenge in telomere research.

The G-strand binding protein Pot1 is one component of shelterin that plays multi-faceted roles in telomere protection and telomere DNA maintenance. Regarding telomere protection, current evidence indicates that mammalian POT1 selectively represses ATR signaling as well as abnormal telomere repair, including HDR and alternative NHEJ (a-NHEJ) [7–11]. However, in fission yeast, the loss of *Pot1* leads to rapid telomere loss, culminating in the

circularization of chromosomes through NHEJ [12,13]. Why the loss of Pot1 results in disparate outcomes is not well understood. POT1 orthologs have also been implicated in telomere DNA maintenance. Mouse POT1b (one of the two POT1 homologs in this organism) interacts with the cognate CTC1-STN1-TEN1 complex (CST) to promote telomere lagging strand synthesis and maturation [14]. CST is another widely conserved telomere complex that promotes telomere maintenance in parallel with shelterin by regulating primase-Pol α [15–17]. Human POT1 mutations that disrupt the POT1-CST interaction also trigger telomere replication defects [18].

An attractive model for dissecting the interplay between telomere proteins and DNA repair factors is the yeast-like basidiomycete fungus *Ustilago maydis*, originally developed by Robin Holliday to investigate recombinational repair [19]. In comparison to budding and fission yeasts, *U. maydis* bears greater resemblances to mammalian cells with respect to the telomere and HDR repair machinery, suggesting that it may be a more accurate model for mammalian telomere regulation [20,21]. *U. maydis* also harbors the canonical 6-bp telomere repeat tracts ([TTAGGG]$_n$/[CCCTAA]$_n$), which are ~500 bp long at each chromosome end [22]. Indeed, we showed that several repair proteins in *U. maydis* promote telomere replication and telomere recombination just like their mammalian orthologs [23–25]. Accordingly, we have taken advantage of *U. maydis* to examine the mechanisms and regulation of repair proteins at telomeres [23–26]. In one of the published studies, we show that the two double strand telomere binding proteins in *U. maydis*, named Tay1 and Trf2, play distinct roles in telomere maintenance and protection. Tay1, in particular, physically interacts with the Blm helicase and acts in conjunction with the latter to promote telomere replication, providing one illustration of the significance of telomere-repair protein interactions [27].

Herein we report our investigation of *U. maydis* Pot1, the conserved and essential G-strand binding protein. Compared to the well characterized mammalian and fission yeast Pot1 orthologs, *Um*Pot1 harbors an extra N-terminal OB fold domain (OB-N), which was recently shown to be conserved in most metazoan Pot1 proteins [28]. Functional analysis in *U. maydis* revealed multi-faceted, Pot1-dependent effects of DNA repair factors on telomere regulation. Specifically, we showed that Pot1 binds directly to Rad51 and regulates the strand exchange activity of Rad51. Deleting the OB-N domain, which is partly responsible for Rad51-binding, caused telomere shortening, suggesting that Pot1 positively regulates the telomere replication function of Rad51. However, when Pot1 is depleted through transcriptional repression and telomeres become deprotected, Rad51 and Brh2 act to instigate aberrant repair and the production of abnormal telomere structures, including high levels of telomere repeat-containing non-coding RNAs (i.e., TERRA and its complement). Thus, Pot1 also negatively regulates the activities of Rad51 and Brh2 to suppress telomere instability. Together, these findings illuminate the evolution of Pot1 and demonstrate how a single telomere protein can regulate the HDR machinery in disparate manners in the contexts of normal and deprotected telomeres.

## Results

### *Um*Pot1s in Basidiomycetes harbor an extra N-terminal OB fold that exhibits no nucleic acid-binding activity

Like the well characterized fission yeast and primate orthologs, *Um*Pot1 possesses three conserved OB fold domains (OB1, OB2 and OB3) and exhibits high affinity binding to the canonical G-strand repeats [26]. However, *Um*Pot1 also contains a long N-terminal extension (~ 350 amino acids before OB1) of unknown function (Fig 1A). To gauge the extent of conservation for this region, we used BLASTp or PSI-BLAST to identify additional fungal Pot1 homologs both within and beyond the Basidiomycota phylum. Notably, the majority of basidiomycete

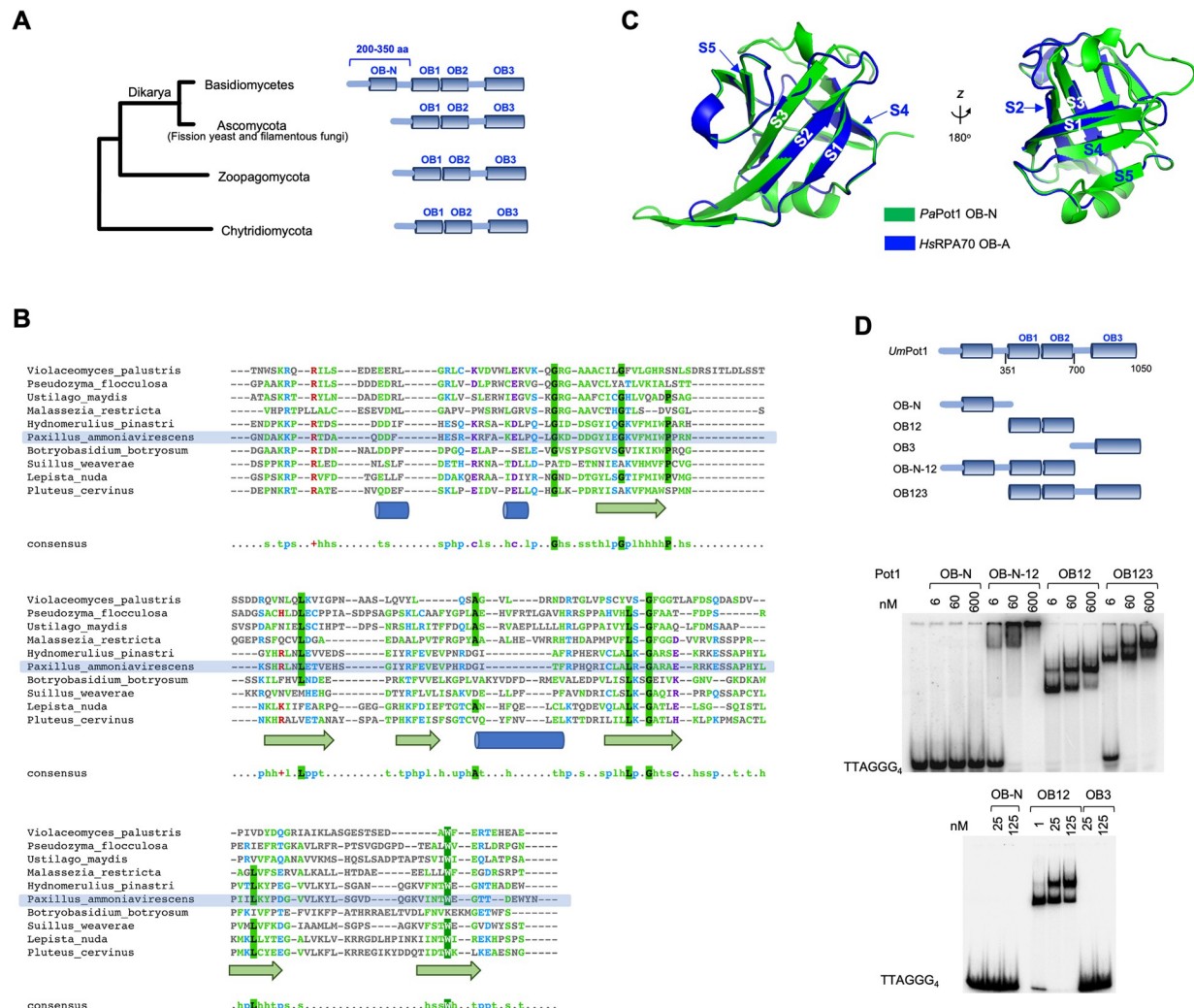

**Fig 1. *U maydis* Pot1 and other basidiomycetes harbor an extra N-terminal OB fold domain. A.** Domain organization of Pot1 homologs from major fungal branches. **B.** Multiple sequence alignment of the N-terminal region of Basidiomycete Pot1 homologs. The alignment and secondary structure predictions were generated using PROMALS3D and displayed using MView. **C.** A homology model of the putative OB-N domain from *Paxillus ammoniavirescens* (*Pa*Pot1) (generated from SWISS-MODEL) is superimposed on the OB-A domain of human RPA70. Note that the five beta strands (labeled as S1 to S5) in the two structures are spatially well aligned. **D.** EMSA analysis of the DNA-binding activities of *Um*Pot1 truncation variants. The domains included in each variant are illustrated at the top and the binding assays shown at the bottom. 5'-labeled G4 oligonucleotide was used as the probe.

Pot1s contain an equivalent N-terminal extension (~200–350 amino acids (a.a.)) before OB1 (Fig 1A). Sequence alignment coupled with secondary structure prediction revealed a moderately conserved 120 a.a. domain exhibiting structural features of an OB-fold, i.e., five β strands with an intervening α helix (Fig 1B) [29]. Thus, most basidiomycete Pot1s appear to harbor four OB-fold domains (designated OB-N, OB1, OB2 and OB3). Interestingly, Myler et al. recently showed that this is true for the majority of metazoan POT1s as well [28]. In support of common ancestry, we found that the two groups of OB-Ns from basidiomycetes and metazoan Pot1s align well to each other (S1A Fig). In addition, homology modeling of a basidiomycete Pot1 OB-N (from *Pa*Pot1) yielded a structure that is superimposable on the OB-A domain from human RPA70 (Fig 1C). Similarities between metazoan and basidiomycete Pot1s are further underscored by the conservation of zinc-binding motifs in the OB3 of these homologs

(S1B and S1C Fig). Interestingly, in contrast to basidiomycetes, the Pot1 homologs in other fungal phyla (e.g., Ascomycota, Chytridiomycota, Zoopagomycota, etc.) contain just three OB folds (Fig 1A). Because phylogenomic analysis indicates that Chytridiomycota and Zoopago-mycota represent deeper fungal lineages than Basidiomycota, it is possible that OB-N might have been independently lost in multiple fungal branches [30], just as it was lost in selected metazoans [28]. Alternatively, basidiomycete Pot1 might have acquired OB-N through the duplication of another OB fold in the protein, such OB1. However, phylogenetic comparison suggests that the OB-N of basidiomycetes is more closely related to the OB-N of metazoans than to the OB1 of basidiomycetes (S2A Fig). Likewise, the same analysis failed to uncover a strong kinship between the OB-N and OB2 of basidiomycete Pot1 (S2B Fig). In addition, given the identical domain organization for the 4-OB Pot1 and RPA1, we assessed the relationship between the OB-N of Pot1 and RPA70 (S2C Fig). Again, the OB-N of metazoan and basidiomycete Pot1 are more similar to each other than to the OB-N of RPA70. Therefore, we favor the notion of a common origin between metazoan and basidiomycete Pot1 OB-N. However, we cannot rule out the possibility that the OB-N of basidiomycete Pot1 was acquired independently and came to resemble its metazoan counterpart through convergent evolution.

In contrast to OB-N, the OB3 domain is evidently well conserved in all fungal phyla, except for the loss of Zinc-binding residues in ascomycetes Pot1 (S1C Fig). We conclude that relative to other fungal Pot1s, basidiomycete Pot1 may be structurally closer to the majority of meta-zoan orthologs.

To assess the contribution of OB-N as well as other domains of *Um*Pot1 to DNA binding, we purified truncation variants of Pot1 and analyzed their G-strand binding activity (Figs 1D and S3A and S3B). Consistent with results for fission yeast and mammalian Pot1 orthologs, we found that high affinity G-strand-binding was mediated exclusively by the OB1 and OB2 of *U. maydis* Pot1. In contrast, neither the OB-N nor OB3 exhibited any appreciable binding activity (Fig 1D). These results suggest that OB-N may mediate its function through protein-protein interactions.

## *Um*Pot1 physically interacts with Rad51 and stimulates the strand exchange activity of Rad51

Since our overall objective in studying *U. maydis* telomeres was to dissect the interplay between telomeres and repair proteins, we screened for potential interactions between Pot1 and DNA repair proteins. Using FLAG-tagged Pot1 in affinity pull down assays, we detected substantial binding of Pot1 to purified Rad51, but not Brh2 (Fig 2A). This binding is probably direct (i.e., not through contaminating nucleic acids) because the addition of benzonase had no effect (Fig 2B). We then examined the Rad51-binding activities of different domains of Pot1 and detected significant activity for OB-N and OB12 of Pot1, but minimal binding for OB3 (S3C Fig). A Pot1 fragment that spans the N-terminus and OB12 showed stronger binding to Rad51 than either domain alone, further supporting the contributions of both regions to Rad51 interaction.

To determine if Pot1 can modulate the recombinase activity of Rad51, we employed an oligonucleotide-based strand exchange assay [31,32]. As expected, Rad51 alone catalyzed strand exchange between the duplex and ssDNA substrates in an ATP-dependent manner (Fig 2C). Interestingly, while Pot1 alone exhibited no strand exchange activity, it stimulated strand exchange by ~ 2-fold in the presence of Rad51 (Fig 2D). By contrast, the duplex telomere protein Tay1, which does not bind Rad51 [27], had no effect on strand exchange either in the absence or presence of Rad51 (Fig 2D). We further examined the ability of Pot1 truncation derivatives to stimulate strand exchange, and found that both the N-terminus and OB12

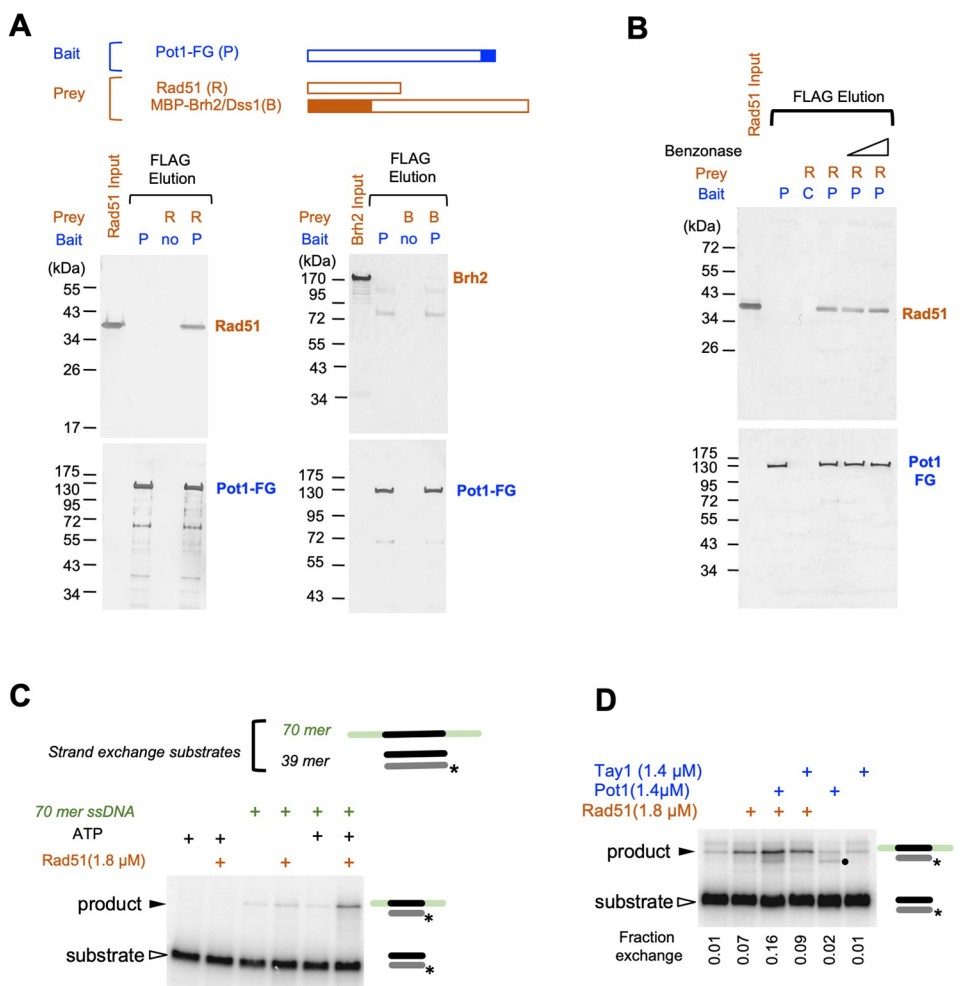

**Fig 2.** ***U maydis* Pot1 binds to Rad51 and regulate the strand-exchange activity of Rad51. A.** Anti-FLAG agarose resin was pre-bound with Pot1-FLAG and tested for interaction with purified Rad51 or MBP-Brh2/Dss1 complex. The bound proteins were eluted with $FLAG_3$ peptides and the levels of Rad51, Brh2 and Pot1 in the eluates analyzed using anti-Rad51, anti-MBP, and anti-FLAG antibodies. **B.** The interaction between Pot1-FLAG and Rad51 was analyzed in the absence or presence of Benzonase. **C.** The ability of Rad51 to mediate strand exchange between a 70-mer ssDNA and a labeled 39-mer dsDNA was analyzed in the absence or presence of ATP. The substrates and products are marked by an open and a closed arrowhead, respectively. Asterisks designate the positions of the 5' $^{32}$P label. **D.** The effects of Pot1 and Tay1 on strand exchange were analyzed in the absence or presence of Rad51. The substrates and products are marked by an open and a closed arrowhead, respectively. Asterisks designate the positions of the 5' $^{32}$P label. An additional band is detected in Pot1-containing reactions (marked by a filled circle). While the identity of this is unclear, it is not Rad51-dependent and unlikely to be relevant to the strand-exchange reaction.

domains were required for stimulation (S3D Fig). Together, these results implicate direct Pot1-Rad51 interaction in the stimulatory effect of Pot1. It is worth noting that the substrates in the strand exchange assays do not harbor telomere repeats (S2 Table) and are not expected to bind Pot1, again suggesting that the effect of Pot1 is mediated through protein-protein interaction rather than DNA-binding.

## *Um*Pot1 promotes telomere maintenance

The Pot1-Rad51 interaction suggests that the telomere maintenance function of Rad51 [24,26] might be regulated by Pot1. To test this idea, we sought to determine the consequence of

disrupting this interaction. Because mutating the OB12 domain will likely abrogate DNA-binding and trigger telomere deprotection (making it difficult to interpret the phenotypes), we deleted just the OB-N of Pot1 (*pot1ΔN*) and used this allele to replace genomic *pot1* through homologous recombination. The *pot1ΔN* mutant did not exhibit obvious growth defect or UV sensitivity (Fig 3A), suggesting that telomere protection is intact and Pot1 does not affect the genome-wide repair function of Rad51. In addition, *pot1ΔN* did not exhibit growth defects or evidence of senescence even following extensive passage on plates (Fig 3B). We then examined telomere maintenance of *pot1ΔN* using Southern analysis of telomere restriction fragments (TRF), and detected a moderate telomere length reduction in later passages of the mutant in comparison to the wild type strain (Fig 3C, streaks 6 and 10). To obtain a more quantitative assessment of telomere lengths, we used STELA (single telomere length analysis) to character-ize the subpopulations of telomeres that harbor the UT4 subtelomere sequence [22]. Consis-tent with Southern analysis of TRF lengths, the STELA products in *pot1ΔN* were ~ 200 bp shorter than wild type at streak 10, supporting the notion that Pot1 promotes telomere mainte-nance (Fig 3D). To further test the notion that Pot1-Rad51 interaction underpins the replica-tion function of Pot1, we compared the telomeres of the *rad51Δ pot1ΔN* double mutant to those of *rad51Δ* and *pot1ΔN* single mutants (S4A Fig). No significant differences were observed, supporting the idea that Rad51 acts in the same pathway as the OB-N of Pot1 to pro-mote telomere maintenance. We also explored the effect of replication stress on the telomere phenotype of *pot1ΔN*. However, even low concentrations of hydroxyurea induced substantial telomere length heterogeneity in the wild type strain and there was no obvious exacerbation of telomere defects in *pot1ΔN* (S4B Fig).

## Transcriptional repression of *Um*pot1 triggers severe growth defects and multiple telomere aberrations

We next interrogated the telomere protection function of *U. maydis pot1*. Because deleting this gene is likely to be lethal, we generated a conditional mutant (*pot1^crg1^*) by replacing the endog-enous *pot1* promoter with that from *crg1*, which supports elevated transcription in arabinose-containing media (YPA) but is strongly repressed in glucose-containing media (YPD). To assess the OB-N of Pot1 in telomere protection, we also substituted the *pot1ΔN* promoter with the *crg1* promoter (*pot1ΔN^crg1^*). Both mutants exhibited normal growth in YPA, but suffered dramatic growth defects in YPD, consistent with telomere deprotection (Figs 4A and S5A and S5B). Indeed, both mutants became growth arrested in glucose, with *pot1ΔN^crg1^* yielding espe-cially small colonies consistent with accelerated arrest. Analysis of cell morphology revealed cells of varying sizes and shapes (S5 Fig), suggesting the arrest can occur in different stages of the cell cycle. This makes *pot1*-deficient cells distinct from several other telomere protein mutants such as *ku70* and *trf2*-deficient cells that show predominantly elongated morpholo-gies consistent with G2/M arrest [23,27].

Analysis of telomere repeat DNAs revealed a multiplicity of defects in both the *pot1^crg1^* and *pot1ΔN^crg1^* mutants grown in YPD, but with some notable differences (Fig 4). First, while the lengths of telomeres (as determined by TRF Southern analysis) became quite heterogenous in both mutants, the overall telomere content was substantially higher in *pot1ΔN^crg1^* (Fig 4B). The heterogeneous telomere fragments of *pot1^crg1^* were detected in the absence of *Pst*I digestion, indicating that they exist as extrachromosomal telomere repeat (ECTR) (S5C Fig). Second, while C-circles (a marker of telomere recombination) were found to be elevated in both mutants, the *pot1ΔN^crg1^* mutant also harbored high levels of G-circles (Fig 4C). Third, while *pot1^crg1^* exhibited a modest increase in ssDNA on the C-strand (~10 fold), *pot1ΔN^crg1^* showed instead extremely high levels of ssDNA on the G-strand (Fig 4D and 4E). Interestingly, the G-

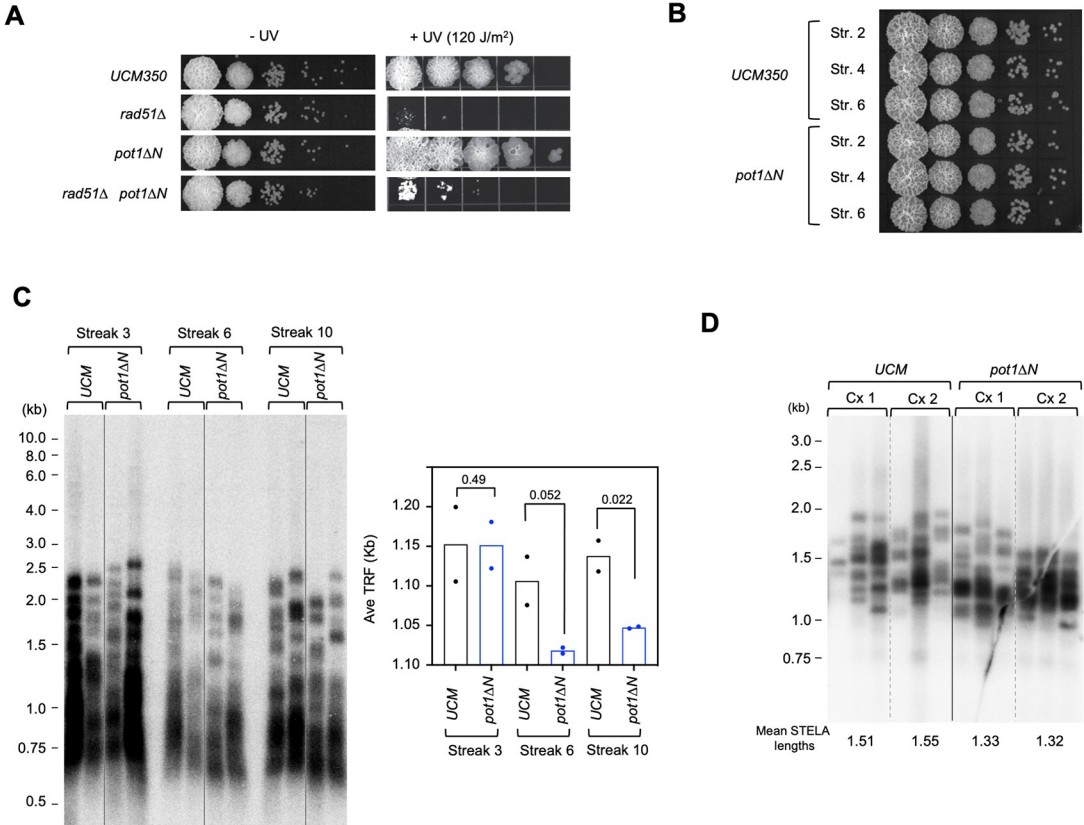

**Fig 3. Pot1 N-terminal deletion impairs telomere maintenance. A.** Serial dilutions of the indicated *U. maydis* strains were directly spotted onto a YPD plate or first subjected to UV irradiation prior to spotting. The plates were incubated for 2 days (-UV) or 3 days (+UV) and the results photographed. **B.** Serial dilutions of cultures of UCM350 and *pot1ΔN* that had been passaged for the indicated numbers of streaks were spotted onto a YPD plate and incubated for 2 days. **C.** Chromosomal DNAs from two independently propagated clonal cultures of UCM350 and *pot1ΔN* were subjected to TRF Southern analysis at the indicated passages (i.e., Streak 3, 6 and 10). Mean TRF lengths from the pairs of assays were calculated and plotted at the bottom. P-values were calculated using two-tailed Student's t tests and displayed. **D.** Chromosomal DNAs from two independently propagated clonal cultures of UCM350 and *pot1ΔN* strains were subjected to STELA analysis at Streak 10. Mean STELA lengths were calculated from 3 independent assays of each DNA sample and shown at the bottom.

strand and C-strand ssDNA each showed some sensitivity to both 3'-overhang and 5'-overhang nucleases (i.e., Exonuclease I and RecJ$_f$) (S5D Fig). This can only be explained by ssDNA on linear ECTRs, which is consistent with the high levels of ECTRs detected in Southern analysis. Together, these observations suggest that loss of *pot1* triggers high levels of both telomere recombination and resection, leading to increases in both ds and ss telomere DNA, with much of the DNA in the form of ECTR. In addition, the differences between the telomere aberrations in *pot1$^{crg1}$* and *pot1ΔN$^{crg1}$* mutants suggest that even at low levels, the residual Pot1 can still alter aberrant DNA repair reaction at telomeres. For example, if Pot1 OB-N interacts with DNA repair factors such as Rad51 to influence DNA processing, the absence of this domain in *pot1ΔN$^{crg1}$* could result in differential generation of ssDNA. In support of residual Pot1 expression, RT-qPCR analysis revealed significant *pot1* RNA in the *pot1$^{crg1}$* mutant grown in YPD, at ~1/2 to 1/3 of the normal *pot1* RNA level in the parental strain (S5E Fig). In contrast to the drastic increases in abnormal structures associated with telomere recombination and resection, PCR analysis of telomere-telomere fusions revealed only low levels of fusions in the *pot1$^{crg1}$* and *pot1ΔN$^{crg1}$* mutants grown either in YPA or YPD (S6 Fig). Hence, non-homologous end joining does not appear to be strongly activated by *pot1* deficiency.

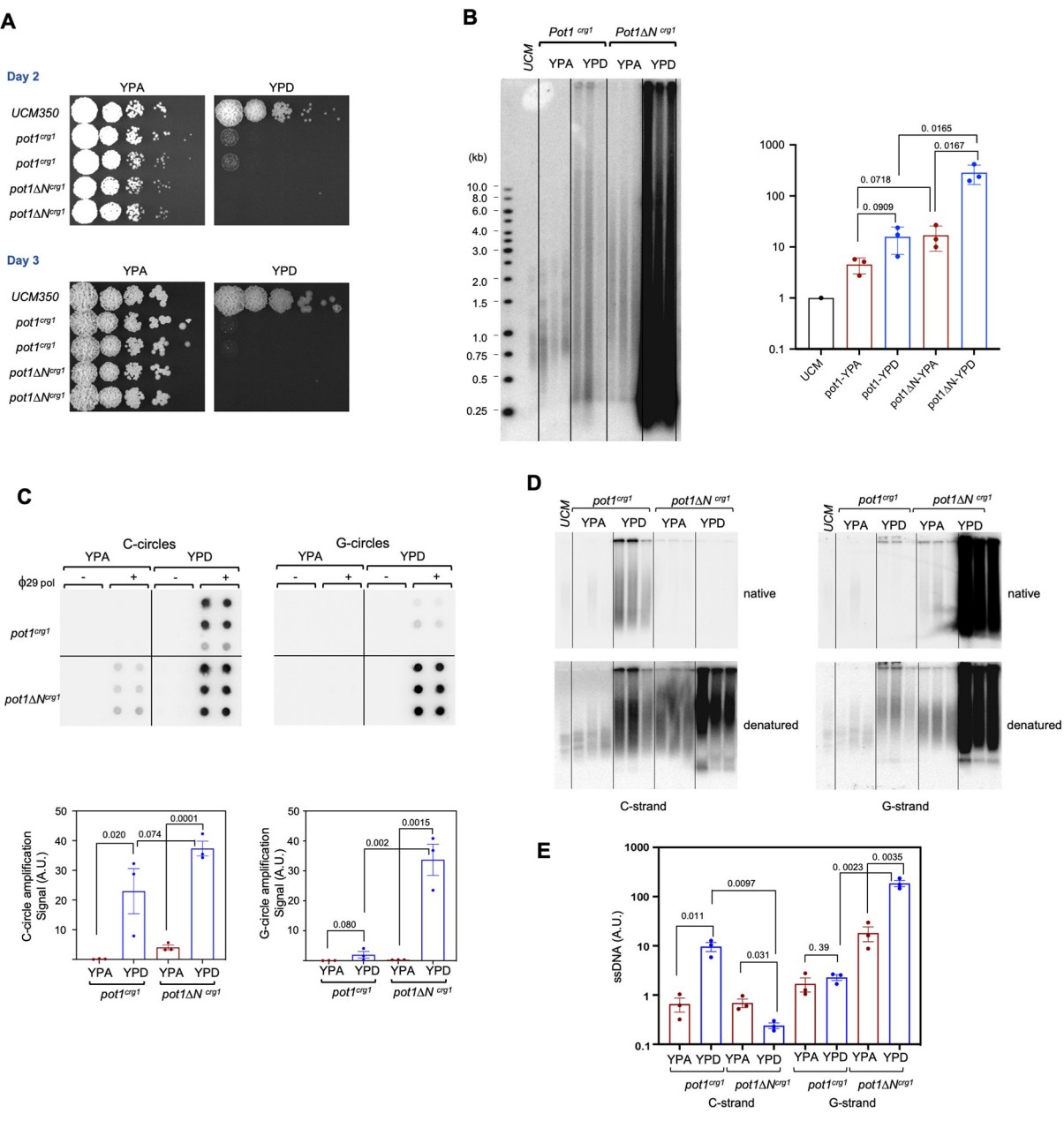

**Fig 4. *pot1* deficiency triggers growth arrest and multiple telomere aberrations. A.** Serial dilutions of the indicated strains were grown on YPA (*crg1* promoter on) and YPD (*crg1* promoter off) plates for 2 and 3 days, and the results photographed. **B.** Chromosomal DNAs from three independently generated *pot1^crg1^* and *pot1ΔN^crg1^* strains grown in YPA and YPD were subjected to telomere restriction fragment analysis. The overall telomere contents of the samples were quantified and plotted on the right. P-values were calculated using two-tailed Student's t tests. **C.** Chromosomal DNAs from three independently generated *pot1^crg1^* and *pot1ΔN^crg1^* strains grown in YPA and YPD were subjected to C-circle and G-circle assays. The relative levels of C- and G-circles from the three mutants were quantified and plotted at the bottom. P-values were calculated using two-tailed Student's t tests. **D.** Chromosomal DNAs from three independently generated *pot1^crg1^* and *pot1ΔN^crg1^* strains grown in YPA and YPD were subjected to in-gel hybridization analysis of ssDNA on both the G- and the C-strand. **E.** The levels of ssDNA from the in-gel hybridization assays were quantified (N = 3) and plotted. P-values were calculated using two-tailed Student's t test.

Interestingly, even when grown in YPA, *pot1ΔN^crg1^* exhibited greater telomere length heterogeneity and a modest increase in C-circle levels (Fig 4A and 4B), suggesting mild telomere de-protection. Because this phenotype was not observed in *pot1ΔN*, it was presumably due to

the over-expression of this truncated allele. While the underlying mechanism is unclear, it is again consistent with the notion that the Pot1ΔN may interact differently with DNA repair factors than Pot1. While Pot1ΔN could retain some interaction with Rad51 through its OB1 and OB2 domains, the overall affinity is likely to be weaker than full length Pot1. There could also be as yet unidentified interactions between Pot1 and other DNA repair factors that are altered by removal of OB-N. High levels of Pot1ΔN may therefore cause an imbalance of DNA repair factors at telomeres to trigger telomere deprotection.

## The growth and telomere defects of *pot1*-deficient *U. maydis* are greatly suppressed by deletions of *rad51* and *brh2*

Recent studies in mammalian cells suggest that *pot1* plays an especially prominent role in suppressing homology-directed repair at telomeres [7,9]. However, this function does not appear to be well conserved in fission yeast [13]. To examine the roles of HDR factors in *pot1*-deficient *U. maydis*, we analyzed the phenotypes of the *pot1^{crg1} rad51Δ* and *pot1^{crg1} brh2Δ* double mutants. Remarkably, both double mutants showed substantially improved growth and manifested more normal morphology in comparison to *pot1^{crg}* when grown in YPD (S7 Fig). Thus, Rad51 and Brh2 were evidently required for the toxicity and growth arrest triggered by *pot1* deficiency, i.e., by mediating aberrant repair. Indeed, analysis of telomere repeat DNAs in the double mutants revealed substantial albeit incomplete suppression of the telomere abnormalities observed in *pot1^{crg1}* when the strains are grown in YPD (Fig 5). For example, both the abnormally long and short telomere fragments detected in *pot1^{crg1}* are greatly reduced (but not eliminated) in *pot1^{crg1} rad51Δ* and *pot1^{crg1} brh2Δ* (Fig 5A). Similarly, the extremely elevated C-strand ssDNA levels are suppressed by ~10 to 20-fold in the double mutants (Fig 5B). Finally, the levels of C-circles (a marker of telomere recombination) in *pot1^{crg1}* are also reduced by more than 10-fold when either *rad51* or *brh2* is deleted (Fig 5C). Hence, the two core HDR factors appear to play a pivotal role in generating aberrant telomere structures in the absence of *pot1*. These results are like those observed in *pot1*-deficient mammalian cells [7], supporting the utility of interrogating the *U. maydis* mutants in dissecting the underlying mechanisms.

It is worth noting that like *rad51Δ* and consistent with our previous report [24], the *brh2Δ* single mutant also exhibited significant telomere shortening. Therefore, even though there is no detectable physical interaction between Pot1 and Brh2 (Fig 2A), the latter may nevertheless cooperate with Rad51 to facilitate telomere maintenance, as has been reported for mammalian cells [33]. Our results suggest that this pathway could be modulated by Pot1-rad51 interaction. One other notable result, from the ssDNA analysis, was the frequent detection of prominent ssDNA signals (both G and C-strand) in the wells for strains bearing either *rad51Δ* or *brh2Δ* mutations regardless of the *pot1* status (Fig 5B, lanes marked with asterisks). This could be due to the accumulation of replication or recombination intermediates, which are known to manifest slow mobility, and is consistent with a role for Rad51 and Brh2 in promoting telomere replication [24].

## Upregulation of telomere-repeat containing RNAs is a cardinal feature of *pot1*-deficient cells

While the observations above implicate Rad51 and Brh2 in promoting aberrant telomere repair in *pot1*-deficient *U. maydis*, the mechanisms by which the various telomere aberrations are produced and how they trigger cell cycle arrest remain incompletely understood. In this regard, it is worth noting that both Pot1 and Rad51 have been linked to the regulation of telomere repeat-containing RNA (TERRA), which is proposed to be a critical determinant of telomere maintenance and stability [34]. It should be noted that TERRA generally refers to RNAs

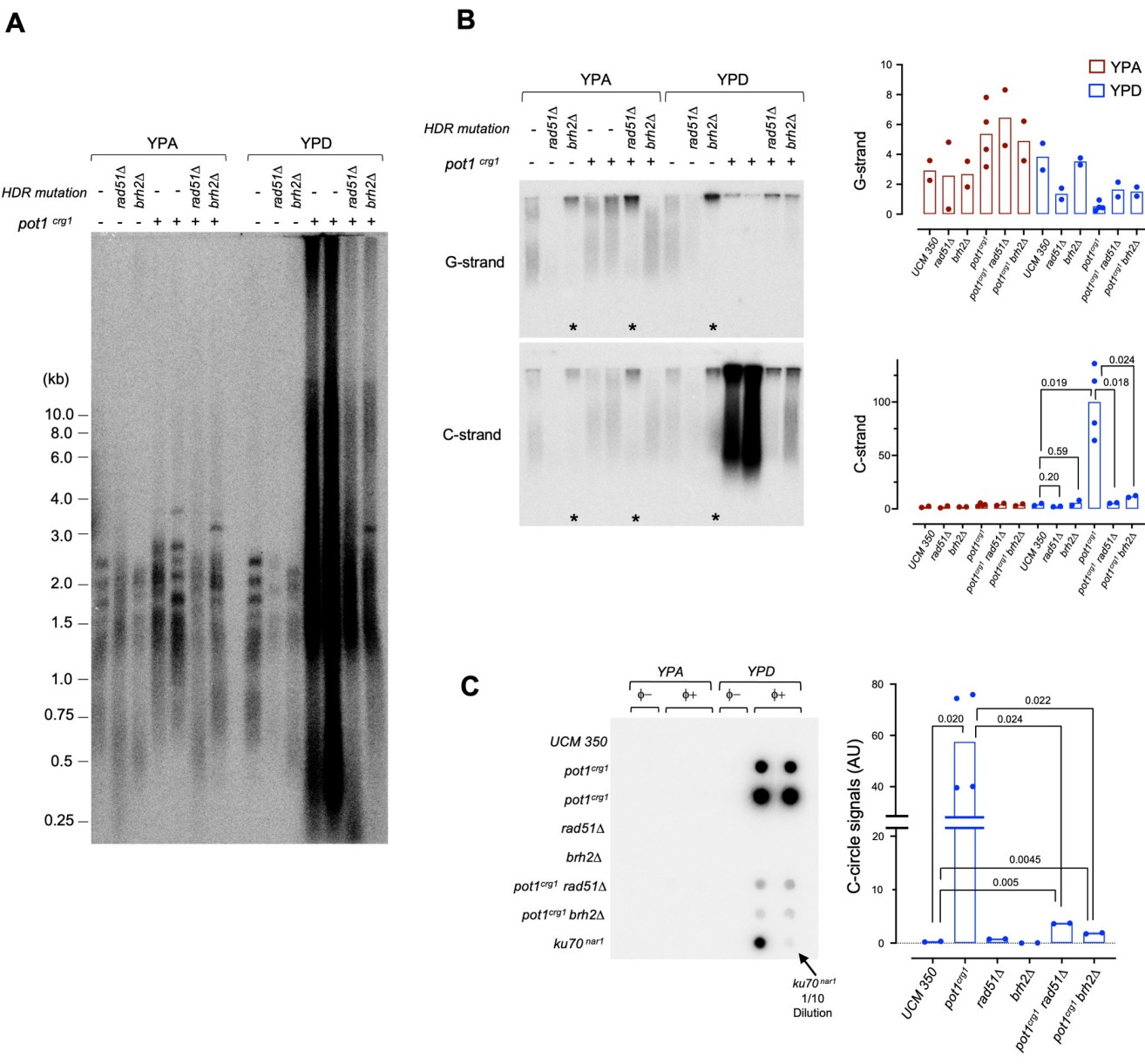

**Fig 5. Telomere aberrations in *pot1*-deficient *U. maydis* are greatly suppressed by *rad51Δ* and *brh2Δ*. A.** Chromosomal DNAs from the indicated wild type and mutant strains grown in YPA and YPD were subjected to telomere restriction fragment analysis. Two independently generated *pot1^crg1* strains were analyzed in all the assays in this figure. **B.** Chromosomal DNAs from the indicated wild type and mutant strains grown in YPA and YPD were subjected to in-gel hybridization analysis of ssDNA on both the G- and the C-strand (left). The levels of ssDNA from two independent sets of assays were quantified and plotted (right). P-values were calculated using two-tailed Student's t tests and displayed. **C.** Chromosomal DNAs from the indicated wild type and mutant strains grown in YPA and YPD were subjected to C-circle assays (left). The relative levels of C-circle signals from these strains were quantified and plotted (right). Data are from two independent sets of assays. P-values were calculated using two-tailed Student's t tests and displayed.

synthesized by Pol II from subtelomeric promoters and elongated into the G-strand repeats; most studies to date addressed the roles of this non-coding RNA species at telomeres [35,36]. However, RNAs that carry the telomere C-strand repeat, named ARIA, have also been reported in *S. pombe* [37,38]. In support of the nexus linking *pot1*, *rad51* and TERRA, POT1 deficiency in human cells was shown to increase TERRA-DNA hybrid formation [7], whereas RAD51 is proposed to directly promote such hybrids through its strand exchange activity [39].

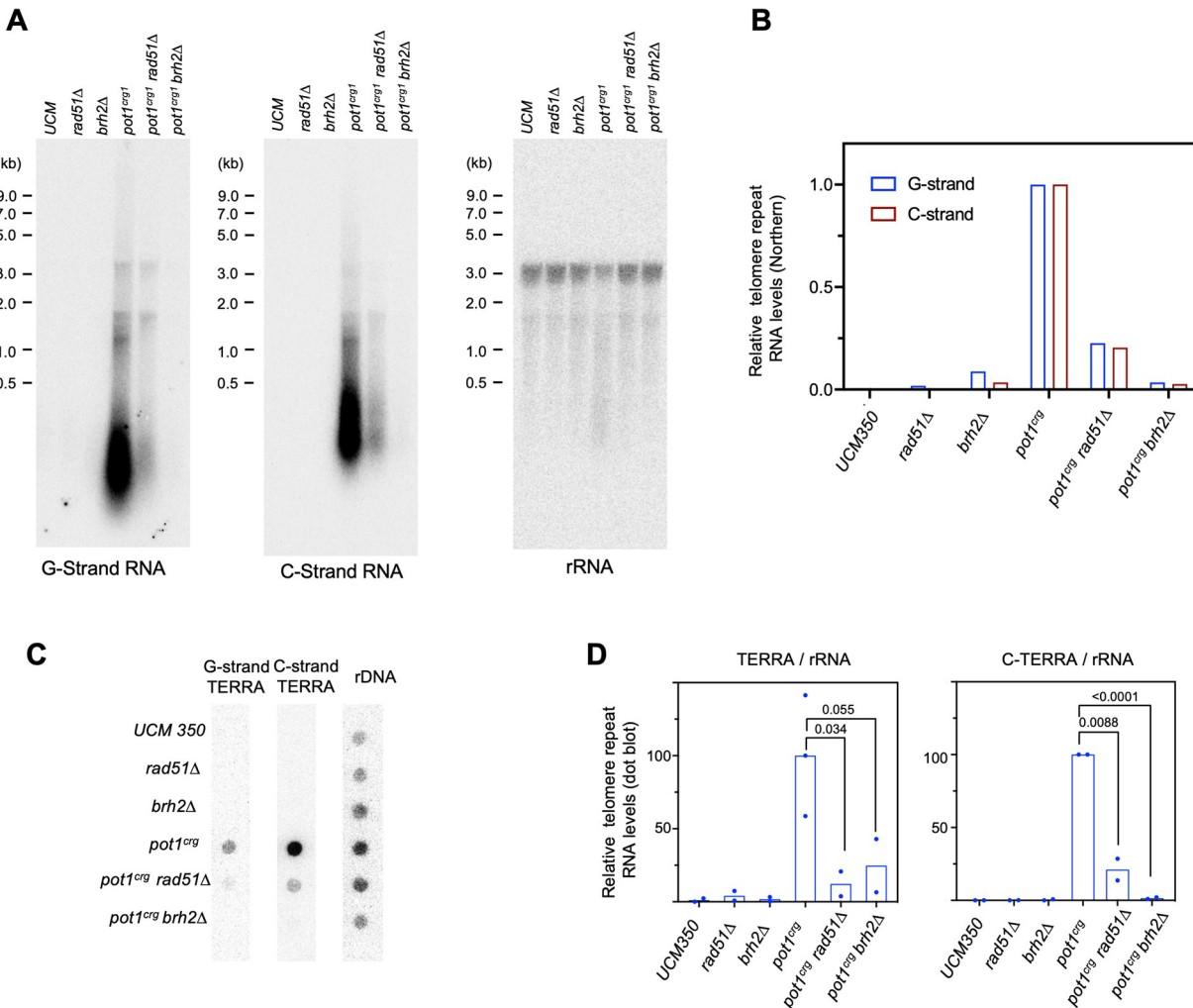

**Fig 6. Telomere repeat-containing RNAs are elevated by *pot1*-deficiency and suppressed by *rad51Δ* and *brh2Δ*. A.** RNAs from the indicated wild type and mutant strains grown in YPD were isolated by acid phenol extraction and subjected to Northern analysis using sequentially probes that detect G-strand repeats, C-strand repeats, and 26S rRNA. **B.** The ratios of G- and C-strand RNA signals to the 26 rRNA signals from Northern analysis were calculated, and then normalized against the ratio of the *pot1^crg1* mutant and plotted. The normalized values for the *pot1^crg1* G- and C-strand RNA levels were each set to 100. **C.** RNAs from the indicated wild type and mutant strains grown in YPD were isolated by acid phenol extraction, treated with Turbo DNase (Thermo Fisher Inc.), and subjected to dot blot analysis using sequentially probes that detect the G-strand repeats, C-strand repeats, and 26S rRNA. **D.** The ratios of G- and C-strand signals to the 26 rRNA signals from dot blot analysis (n = 2 or 3) were calculated, and then normalized against the ratio of the *pot1^crg1* mutant and plotted. P-values were calculated using two-tailed Student's t tests and displayed.

These precedents argue that TERRA may participate in the pathways mediated by *pot1*-deficiency and *rad51/brh2* deletions.

To characterize the potential involvement of telomere-encoded RNA in our mutant strains, we first analyzed the existence of such RNAs by Northern analysis and found little RNA in wild type cells but high levels of transcripts in *pot1*-deficient mutants. Using strand-specific probes, we showed that RNAs corresponding to both G- and C-strands of telomeres were strongly up-regulated in the *pot1* deficient mutant (Fig 6A and 6B). These transcripts were heterogeneous in size but predominantly short (< 1000 nt). RT-PCR analysis confirmed the upregulation of both G- and C-strand RNAs and further suggests that these RNAs include both the subtelomeric region and telomere repeats (S8 Fig). For example, the UT6 subtelomeric region

can be amplified using cDNA generated by the CCCTAA$_4$ primer, consistent with the existence of G-strand TERRA that spans the subtelomeric-telomeric junction (S8 Fig). Likewise, heterogeneous PCR products were generated when C-strand RNA was reverse transcribed with a UT6 primer and then amplified using forward and reverse primers that anneal to UT6 and telomere repeats, respectively (S8 Fig). This again implies the existence of C-strand RNAs that span the subtelomeric-telomeric junction. Notably, our data do not exclude the presence of C-strand RNAs that comprise exclusively of just the telomere repeats, called ARIA [38]. However, to avoid confusion, we will refer to our C-strand subtelomere-telomere RNAs as C-TERRA. In addition, our data are consistent with the existence of C-strand RNAs that comprise exclusively of subtelomere sequence, named ARRET (S8 Fig). Altogether, the Northern and RT-PCR analyses demonstrate that abnormally high levels of both G- and C-strand RNAs are present in *pot1*-deficient cells, due either to increased transcription or to increased RNA stability. Although the end-point PCR assays could not provide quantitative assessment of RNA levels, the relative signals suggest that C-TERRA was present at higher levels in the mutant (see below for additional evidence).

We next compared the levels of telomere repeat RNAs in the *pot1* single and double mutants by both Northern and dot blots (Fig 6A–6D). Both analyses showed that while there were drastic increases in TERRA and C-TERRA (by > 20-fold) in the *pot1* single mutant, these increases were strongly suppressed by *rad51* and *brh2* deletions (by > 5-fold). Thus, aberrant transcription of telomere repeats from both G- and C-strands is also regulated by HDR factors in *pot1*-deficient *U. maydis*. Lastly, we used specific nuclease treatments to assess the potential existence of RNA-DNA hybrids in the acid phenol-extracted nucleic acid preparations (prior to DNase I treatment), which should be enriched for both RNA and RNA-DNA hybrid (S9 Fig). Interestingly, we found that the G-strand hybridization signals in the *pot1$^{crg1}$*, *pot1$^{crg1}$ rad51Δ*, and *pot1$^{crg1}$ brh2Δ* mutants were all substantially reduced by pre-treatment with DNase I, implying that the G-strand DNA: C-TERRA hybrid (hybrid I) represents a substantial fraction of the signals. The signals were also partially sensitive to RNase H cleavage, indicating the presence of the alternative hybrid (C-strand DNA: TERRA; hybrid II). In contrast, the C-strand signals in the dot blot assays were mostly insensitive to either nuclease, suggesting that they were predominantly in the form of ssRNA or RNA-RNA hybrid. Notably, the C-strand RNA in hybrid I should be degraded by RNase H and should render the C-strand dot blot signal RNase H-sensitive if hybrid I is abundant relative to free C-strand RNAs. This not being the case, we surmise that telomere C-strand RNA may be substantially more abundant than G-strand TERRA in the *pot1$^{crg1}$* mutant, implying differences between these two RNA species with respect to their production or stability.

## Discussion

There is growing evidence that HDR factors play multi-faceted and seemingly contradictory roles at telomeres. In the context of normal telomere protection, HDR factors facilitate telomere maintenance by stabilizing stalled forks or promoting repair of truncated telomeres [33]. However, when protection is compromised, HDR factors are often responsible for triggering further aberrations and exacerbating genomic instability [9,10]. How telomere proteins interact physically and functionally with DNA repair factors to achieve context-dependent regulation remains a key area of investigation. In this report, we focus on the interplay between the G-strand binding protein Pot1 and the core HDR factors Rad51 and Brh2/BRCA2 using a fungal model that recapitulates crucial aspects of mammalian telomere regulation. We showed that Pot1 negatively regulates Rad51 and Brh2 to prevent aberrant repair that results in loss of cell viability, but also positively regulates Rad51 activity to promote telomere maintenance,

providing an interesting case study for how a single telomere protein can achieve dual regulation of the same DNA repair factor. We also show that up-regulation of aberrant transcription from both G- and C-strand of telomeres is a consistent consequence of telomere deprotection. In addition, we discovered an extra N-terminal OB-fold in the Pot1 orthologs of basidiomycetes, which is absent in other fungal phyla but shared by the majority of metazoans. The mechanistic and evolutionary implications of these observations are discussed below.

## *U. maydis* Pot1 suppresses the toxic activities of core HDR factors at telomeres

Previous analysis of *S. pombe* Pot1 (*Sp*Pot1) suggests that fungal Pot1 may act differently than the mammalian orthologs in telomere protection. To wit, rather than aberrant recombination, loss or inactivation of *Sp*Pot1 leads to rapid telomere resection/loss followed by frequent circularization of chromosomes to stabilize the genome and sustain cell viability [12,13]. In contrast, we showed that the phenotypes of *Um*Pot1-deficient cells are very similar to those of comparable mammalian cells, suggesting that suppression of HDR activities may be a widely conserved function of Pot1 in evolution. Importantly, the loss of viability triggered by *U. maydis pot1*-deficiency is almost completely suppressed by *rad51Δ* or *brh2Δ*, confirming the toxicity of HDR factors in the context of telomere dysfunction. This toxicity is at least somewhat specific given that *rad51Δ* fails to ameliorate the telomere toxicity triggered by *ku70* deficiency [23]. Instead, the *ku70* deficiency-induced abnormalities are completely suppressed by concurrent *blm* deletion [25]. Thus, there is a clear division of labor among telomere proteins in managing aberrant repair. The mechanistic underpinnings for this remain unclear and demand further investigation.

Viewed from another perspective, it is striking that in *U. maydis*, *ku70* and *pot1* become dispensable for cell viability when specific DNA repair factors are deleted. Such examples of telomere protein dispensability in the context of DNA repair deficiency reify the concept that telomeres have evolved to restrain aberrant repair. It is worth noting that in mammalian cells, there are few examples where the loss of DNA repair factors has been shown to restore the viability of cells missing a key telomere protein. This is probably because mammalian DNA repair factors are often essential for viability in normal mitotic cells. The reduced dependency of *U. maydis* on DNA repair makes it possible to firmly establish the causal connections between telomere deprotection, aberrant telomere DNA repair, and loss of cell viability.

## *U. maydis* Pot1 promotes telomere maintenance by enhancing the function of Rad51

POT1 has been proposed to promote telomere maintenance in mammalian cells through multiple mechanisms. First, POT1 unwinds G-quadruplexes *in vitro*, and this biochemical activity is thought to facilitate telomerase activity and telomere replication [40,41]. Second, POT1 binds to and acts in conjunction with the CST complex to facilitate telomere replication and C-strand fill-in synthesis [14,18]. Our analysis of *U. maydis* Pot1 suggests an additional, non-mutually exclusive mechanism, namely through the regulation of Rad51 activity. Notably, while we showed that Pot1 can stimulate the strand exchange activity of Rad51 through direct protein-protein interaction, how this *in vitro* regulation reflects the *in vivo* mechanism of Rad51 at telomeres remains unclear. If Pot1 enhances the formation or stability Rad51 filaments, this could promote complete replication by stabilizing stalled forks. However, it is also clear that rampant strand exchange by Rad51 at telomeres is toxic to the cell and is restrained by Pot1 (see the section above). Thus, it is possible that Pot1-Rad51 interaction may positively regulate some aspects of Rad51 activity while suppressing others in a context dependent

manner, e.g., through modulation by other telomere and DNA-repair factors. More studies will be necessary to disentangle the molecular basis of Pot1-mediated regulation of Rad51 in *U. maydis*.

Notably, mammalian RAD51 is also implicated in telomere replication [33], but a direct POT1-RAD51 connection has not been reported. Instead, there is substantial evidence that RAD51 is regulated by CST, the complex that acts in parallel with shelterin to promote telomere maintenance. For example, CST associates with Rad51, and CST deficiency impairs the recruitment of RAD51 to genomic sites of replication stress [42]. In addition, CTC1 disease mutations that compromise telomere maintenance often disrupt its association with RAD51 [43]. Whether POT1 acts in conjunction or in parallel with CST to regulate RAD51 function at mammalian telomeres is an interesting question for future analysis. Conversely, it will be worthwhile to determine if *U. maydis* CST subunits regulate Rad51 function. However, it should be noted that only Stn1 and Ten1 homologs are readily discernible in the *U. maydis* genome (just like *S. pombe*), suggesting that some fungal lineages may have undergone significant remodeling of the CST complex [24,44].

## Aberrant transcription of telomere repeats at de-protected telomeres

There is growing evidence that while TERRA is crucial for normal telomere regulation [34,45], it is often aberrantly expressed from damaged telomeres and can promote abnormal telomere metabolism by forming RNA-DNA hybrids or by interacting with a multiplicity of factors [46–49]. Of more direct relevance to the current study, the depletion or knockout of shelterin components in both mammals and fungi has been shown to trigger the production of telomere repeat RNAs as well as RNA-DNA hybrids. For example, depleting human TRF2 leads to rampant up-regulation of TERRA, which in turn alters histone methylation and promotes end-to-end fusion [50]. Similarly, knocking out human POT1 triggers both abnormal recombination as well as the accumulation of telomere RNA-DNA hybrid [7]. Consistent with these precedents, we observed dramatic upregulation of telomere repeat containing RNAs in *pot1*-deficient *U. maydis*. However, both the structure of the RNAs and their relative expression manifest notable features.

One notable feature is the structure of the C-strand RNA. In mammalian cells, this RNA species is evidently expressed at very low levels (if at all) from normal telomeres, and its structure has not been investigated in detail [35,36]. In *S. pombe*, the C-strand RNAs are synthesized to some degree from normal telomeres and are dramatically up-regulated in a number of mutants, including *rap1Δ*, *taz1Δ*, and *poz1Δ* [38,51]. However, unlike C-TERRA in *U. maydis*, which encompasses both subtelomere and telomere repeats, the *S. pombe* C-strand RNAs are reported to comprise just the telomere repeat (ARIA) or subtelomere regions (ARRET). This discrepancy indicates either there are species-specific differences in the biogenesis of C-strand RNA, or the *S. pombe* ARIA and ARRET could arise through post transcriptional processing. An unresolved conundrum concerning C-strand repeat RNA (ARIA or C-TERRA) is that transcription must initiate within the telomere repeat region, which (unlike subtelomeric Pol II promoters responsible for TERRA transcription) does not contain promoter- or initiator-like elements. Thus, an unusual, promoter-independent mode of RNA synthesis is probably involved. A potential precedent for this is provided by a recent report of MRN-dependent transcription by RNA Polymerase II at DSBs [52]. Regardless of the potential mechanisms, our observation suggests that C-strand RNA synthesis may be relatively common at deprotected telomeres and is worthy of further investigation. Conversely, the growing evidence for promiscuous transcription at de-protected telomeres suggests that a key function of telomere proteins is in fact to suppress aberrant transcription.

Another unexpected finding is the greater abundance of C-TERRA relative to TERRA in the *pot1* mutants. As described before, both the RT-PCR and dot blot analyses support the idea that C-TERRA is present at higher levels when *pot1* is depleted (S8 and S9 Figs). Whether this is due to enhanced synthesis or greater stability of the C-strand RNA is unclear. The high levels of C-TERRA may partially account for the formation of hybrid I (G-strand DNA: C-TERRA), which was readily detected in the mutant. Most studies of telomere RNA-DNA hybrid have focused on the roles of hybrid II (C-strand DNA: TERRA) and the associated R-loops, and have implicated these structures in telomere abnormalities. Our identification of hybrid I uncovers yet another structure that could be responsible for exacerbating telomere damage.

## Implications for the evolution of telomere protein complexes

The discoveries that basidiomycete Pot1 has four OB-fold domains and that *Um*Pot1 interacts with Rad51 have evolutionary implications. Based on structural and functional considerations, it has been argued that RPA, CST and POT1-TPP1 may represent ancient paralogs (i.e., RPA1 ≈ CTC1 ≈ POT1 and RPA2 ≈ STN1 ≈ TPP1) and accordingly may share more structural and functional similarities than hitherto realized [17]. Our analysis, coupled with that by Myler et al. [28], indicates that the common ancestor of fungal and metazoan POT1 probably contained four OB-folds and utilized the two central OBs (OB1 and OB2) for high affinity DNA-binding, just like RPA1. This notion reinforces the growing evidence for structural similarities–and hence evolutionary kinship–between these OB-fold-rich complexes (S10 Fig) [53]. Also, as discussed above, the regulation of RAD51 by CST in mammals and its regulation by Pot1 in *U. maydis* underscore functional similarities between CST and POT1-TPP1, albeit in different organisms. Further analysis of RPA, CST and POT1-TPP1 in fungi, metazoan and deeper branches of eukaryotes should allow us to adjudicate the evolutionary relationships between these complexes.

## Materials and methods

### Pot1 homolog identification, alignment, structural prediction, and phylogenetic analysis

Fungal homologs of *Um*Pot1 were identified using BLASTp or PSI-BLAST with appropriate taxonomic restrictions. Similar numbers of homologs that belong to the Basidiomycota, Ascomycota, and Chytridiomycota/Zoopagomycotina lineages were collected for further analysis. The IDs for these homologs are listed in S3 Table. Homologs were aligned using either PSI-Coffee (http://tcoffee.crg.cat/) or PROMALS3D (http://prodata.swmed.edu/promals3d/promals3d.php), and the output displayed using MView (https://www.ebi.ac.uk/Tools/msa/mview/; set to default parameters except coloring by group). The secondary structure predictions associated with the alignments were obtained from the PROMALS3D output and added to the MView displays. Homology modeling was carried out using either PHYRE2 (http://www.sbg.bio.ic.ac.uk/phyre2/html/page.cgi?id=index) or SWISS-MODEL (https://swissmodel.expasy.org/) and the predicted structures displayed using PYMOL. To test the hypothesis that basidiomycete OB-N may have arisen through duplication of OB1, we assessed the similarity of basidiomycete OB1 to both basidiomycete OB-N and metazoan OB-N. Briefly, a multiple sequence alignment between the three groups of OB folds was generated using T-Coffee. The Phylip alignment was then used to infer phylogeny via FastME 2.0 (http://www.atgc-montpellier.fr/fastme/) [54] and the results plotted using FigTree [55]. The same strategy was used to analyze basidiomycete Pot1 OB2 and RPA1 OB-N as possible sources of basidiomycete Pot1 OB-N.

## *Ustilago maydis* strain construction and growth conditions

Standard protocols were employed for the genetic manipulation of *U. maydis* [56–58]. All *U. maydis* strains used in this study were haploid and were derived from the UCM350 background [57,59] and are listed in S1 Table.

The *pot1$^{crg1}$* strain was constructed by integrating a linearized pRU12-*pot1* plasmid containing fragments that correspond to (1) 700 bp of the *pot1* promoter and (2) the first 700 bp of the *pot1* ORF into the *pot1* genomic locus in UCM350. Briefly, the two *pot1* fragments were generated by PCR using appropriate oligos with flanking restriction enzyme sites (S2 Table), and then cloned together in between the *Nde*I and *Eco*RI sites of pRU12 [60]. Correct recombination of *Xba*I treated pRU12-*pot1* into the genomic locus results in the insertion of the *cbx$^R$* marker and the *crg1* promoter precisely adjacent to and upstream of the *pot1* ATG initiation codon. Following transformation, the correct mutant strains (*pot1$^{crg1}$*) were selected on YPA supplemented with carboxin (4 μg/ml) and confirmed by PCR. The *pot1ΔN$^{crg1}$* was generated in identical fashion except that nucleotide (nt) 1126 to 1825 of the *pot1* ORF was used in place of the nt 1 to 700 fragment in the cloning of the pRU12-*pot1ΔN* disruption plasmid. The *pot1ΔN* strain was constructed by integrating linearized pCbx-*pot1ΔN*, a derivative of pRU12-*pot1ΔN* that contains (i) a 5' homology region with nt -1400 to -700 of the *pot1* promoter, and (ii) a 3' homology region with nt -700 to -1 of the *pot1* promoter fused to nt 1226 to 1825 of the ORF. The *rad51Δ* and *brh2Δ* disruption cassettes have been described before [24]. The *pot1$^{crg1}$ rad51Δ* and *pot1$^{crg1}$ brh2Δ* mutants were generated by transforming *pot1$^{crg1}$* with the same *rad51* and *brh2* knockout cassettes as those used for the derivation of the single mutants [24]. The *pot1$^{crg1}$* and *pot1ΔN$^{crg1}$* strains were grown in YPA (1% yeast extract, 2% peptone, 1.5% arabinose) or YPD (1% yeast extract, 2% peptone, 2% dextrose) to induce or repress *pot1* expression, respectively.

For growth analysis of the mutants, liquid cultures of *U. maydis* strains were diluted to OD$_{600}$ of 0.2 (corresponding to ~ 1.5 x 10$^6$ cells/ml). Four microliters of 6-fold serial dilutions of the cultures were then spotted onto YPA or YPD plates and grown at 30°C for 2 to 3 days.

## RT-qPCR analysis of *pot1* RNA levels

For RT-qPCR analysis of *pot1* expressions levels, RNAs were extracted and purified from log-phase *U. maydis* cultures using acid phenol extraction [61]. The samples were treated with TURBO DNase (Thermo Fisher) and then reverse transcribed in 20 μl reactions that contained 1x SSIV Buffer, 12.5 μM primer (Pot1-PCR-3020R), 1 mM dNTP, 5 mM DTT, 20 U NxGen RNase Inhibitor (Lucigen), and 40 U Superscript IV RT (Thermo Fisher). The reaction mixtures were incubated at 55°C for 40 min and then heated at 70°C for 15 min to inactivate the reverse transcriptase. qPCR was carried out in Chai Open qPCR Dual Channel thermocycler using Chai Green Master mix with 8 μM primers (Pot1-PCR-2821F and Pot1-PCR-3020R) and 1 μl of cDNA. Cycling parameters were as follows: 45 cycles of 95°C for 30 sec, 53°C for 20 sec, 72°C for 40 sec.

## Telomere length and structural analyses

Southern analysis of telomere restriction fragments (TRF Southern) was performed using DNA treated with *Pst*I as described previously [24,26]. The blots were hybridized to a cloned restriction fragment containing 82 copies of the TTAGGG repeats. For quantitation, we used the formula 'mean TRF length = Σ (OD$_i$ × L$_i$) / Σ (OD$_i$)' because the majority of TRF length variation stems from differences in subtelomeric lengths [62,63]. The in-gel hybridization assays for detecting ssDNA on the G- and C-strand have also been described [23]. STELA for telomeres bearing UT4 subtelomeric elements was performed essentially as reported before

[22]. To detect telomere-telomere fusions, chromosomal DNA was subjected to PCR using two subtelomeric primers (UT4-F and UT6-F) that are extended by DNA polymerase towards the chromosome ends. The PCR reactions (20 μl) contained 1x Failsafe PreMix H, 0.1 μM each of UT4-F and UT6-F, and 2 U Failsafe polymerase; the thermocycling consisted of 36 cycles of 94°C for 30 sec, 63°C for 30 sec, and 72°C for 2 min. The fusion PCR products were detected by Southern using either a UT4 or UT6 subtelomeric probe [27]. C-circle and G-circles assays were performed as before except that NxGen phi29 DNA Polymerase (Lucigen Corp.) were used for rolling circle amplifications [26].

### Analysis of telomere repeat-containing RNAs

RNAs were purified from log-phase *U. maydis* cultures using acid phenol extraction [61]. For Northern analysis, 3 μg of total RNA were resolved by electrophoresis through a 1.0% agarose/formaldehyde gel [64]. After photography under UV light to visualize ethidium bromide-stained RNAs, the gel contents were transferred to a Hybond-N membrane (GE Healthcare) and hybridization was performed in Church mix using a $^{32}$P-labeled single-strand oligonucleotide complementary to the G-strand repeats ($CCCTAA_8$). Following exposure to a phosphor screen, the blot was stripped and first re-probed for the C-strand telomere repeat RNA using $^{32}$P-labeled $TTAGGG_8$, and subsequently for 26S rRNA using a PCR fragment (amplified with the UmrRNA_26S_121F and UmrRNA_26S_642R primers (S2 Table)). For dot blot analysis, 0.2 μg of RNAs (untreated or pre-treated with various nucleases) in 10 μl were diluted with 30 μl of RNA incubation/denaturation solution (65.7% formamide, 7.7% formaldehyde, 1.3x MOPS buffer), heated at 65°C for 5 min, and then cooled on ices. After adding 40 μl of 20x SSC, the entire samples were applied to Hybond-N using a Bio-Dot Microfiltration Apparatus (Bio-Rad Laboratories, Inc.). After UV crosslinking, the blot was hybridized sequentially to the same probes as those for Northern analysis. For end-point RT-PCR, the RNA samples from acid phenol extraction were first treated with Turbo DNase (Thermo Fisher Inc.). Fifty ng of RNAs were subjected to reverse transcription using Superscript III RT at 55°C for 40 min, followed by heating at 70°C for 15 min to inactivate RT, and then subjected to PCR (33–37 cycles of 95°C for 15 sec, 66°C for 15 sec, and 72°C for 30 sec) with Q5 polymerase (NEB Inc.) and appropriate primers (S2 Table).

### Purification of Pot1-FG and truncation variants

To express full length and deletion mutants of *U. maydis* Pot1, we amplified the *pot1* open reading frame carrying a C-terminal FLAG tag from genomic DNA by PCR amplification using suitable primer pairs (S2 Table). The PCR fragments were cloned into the pSMT3 vector to enable the expression of $His_6$-SUMO-Pot1-FG fusion proteins. BL21 codon plus strains bearing the expression plasmids were grown in LB supplemented with kanamycin and induced by IPTG as previously described [26]. Each fusion protein was purified by (i) Ni-NTA chromatography, (ii) ULP1 cleavage and, and (iii) anti-FLAG affinity chromatography as previously described.

### EMSA assays

For DNA binding assays, purified Pot1-FG was incubated with 1–10 nM $P^{32}$-labeled probe and 0.5 μg/μl BSA in binding buffer (25 mM HEPES-KOH, pH 7.5, 5% glycerol, 3 mM $MgCl_2$, 0.1 mM EDTA, 1 mM DTT) at 22°C for 15 min, and then subjected to electrophoresis at 0.8 V/cm through a 6% non-denaturing polyacrylamide gel in 1x TBE, followed by PhosphorImager analysis.

## Pull down assays for Pot1-Rad51 and Pot1-Brh2 interactions

Pot1-FLAG was immobilized on FLAG affinity resin and incubated with 1 μM purified Rad51 or MBP-Brh2/DSS1 complex at 4˚C in FLAG(65) buffer (50 mM Tris-HCl, pH 7.5, 65 mM NaCl, 10% glycerol, 0.1% NP-40, 2.5 mM $MgCl_2$, 1 mM DTT). Following incubation with mixing on a rotator at 4˚C for 1 h, the beads were washed 4 times with 0.5 ml of the FLAG(100) buffer (same as FLAG(65) except that NaCl is at 100 mM), and then the bound proteins eluted with 60 μl FLAG(150) (same as FLAG(65) except that NaCl is at 150 mM) containing 0.2 mg/ml $FLAG_3$ peptide. The eluates were analyzed by Western Blot using anti-Rad51 or anti-MBP antibodies.

## Rad51 strand exchange assays

The assays were modified from published protocols [31,32] and were performed in 15 μl of SE buffer (25 mM Tris.HCl, pH 7.5, 25 mM KCl, 2.4 mM $MgCl_2$, 1 mM DTT, 2 mM ATP) containing 0.25 μM labeled dsDNA (39-mer with the $P^{32}$-labeled top strand), 0.75 μM unlabeled ssDNA (70-mer complementary to the top strand of the 39-mer), and indicated amounts of Rad51 and Pot1-FG. After 20 min of incubation at 35˚C, the reactions were terminated by the addition of 5 μl stop solution (80 mM EDTA, 4% SDS) containing 0.5 μg/μl proteinase K, and incubated for another 20 min at 35˚C. Following the addition of 4 μl loading dye (40% glycerol, 0.25% bromophenol blue, 0.25% xylene cyanol), the samples were analyzed by electrophoresis at 0.8 V/cm through a 10% polyacrylamide gel in 1x TBE, and visualized by PhosphorImager scanning.

## Supporting information

**S1 Fig. Comparisons of Pot1 homologs in metazoans and fungi. A.** Multiple sequence alignment of the putative OB-N domains from metazoans and basidiomycetes. The alignment and secondary structure predictions were performed using PROMAL3D and displayed using MView. Amino acids were colored by groups. **B.** Multiple sequence alignment of the OB3 domains of Pot1 homologs from Basidiomycota, Chytridiomycota, and Zoopagomycota. **C.** Multiple sequence alignment between the OB3 domains of Pot1 homologs from Basidiomycota and Ascomycota. The putative Zinc-binding residues in the basidiomycete OB3 domains are highlighted.
(TIF)

**S2 Fig. Phylogenetic analysis of Pot1 OB-N in relation to (i) OB1 and OB2 in Pot1 and (ii) OB-N in RPA70. A.** To analyze the relatedness of basidiomycete Pot1 OB1 to the OB-N domains from basidiomycete and metazoan Pot1, we generated a multiple sequence alignment between these three groups of OB folds using T-Coffee. The Phylip alignment was used to infer phylogeny via FastME 2.0 (http://www.atgc-montpellier.fr/fastme/) [54] and the result was plotted using FigTree [55]. **B.** The same analysis was performed for basidiomycete Pot1 OB2 vis-à-vis basidiomycete and metazoan Pot1 OB-N domains. **C.** The same analysis was performed for fungal RPA1 OB-N vis-à-vis basidiomycete and metazoan Pot1 OB-N domains.
(TIF)

**S3 Fig. The Rad51-binding and stimulatory activities of Pot1 truncation variants. A.** The domain structure of Pot1 and the truncation variants analyzed in this study are shown. **B.** Affinity purified Pot1 proteins were analyzed by SDS-PAGE and Coomassie staining. **C.** Purified Rad51 was subjected to pull down analysis using FLAG-tagged Pot1 and truncation derivatives. The eluates were analyzed via Western using anti-Rad51 (for Rad51) and anti-FLAG (for Pot1) antibodies. **D.** The effects of Pot1 truncations on the strand exchange activity of

Rad51 were analyzed using oligonucleotide substrates.
(TIF)

**S4 Fig. Genetic interaction between *pot1ΔN* and *rad51Δ*; effects of hydroxyurea on telomeres. A.** Chromosomal DNAs from three independently propagated cultures of each strain were isolated after ~100 generations of growth (4 streaks) and subjected to telomere restriction fragment analysis. The blot is displayed on the left and the average TRF lengths plotted on the right. **B.** The UCM350 and *pot1ΔN* strains were grown in YPD liquid cultures with the indicated concentrations of hydroxyurea and for the specified numbers of days. Chromosomal DNAs were isolated and subjected to telomere restriction fragment analysis.
(TIF)

**S5 Fig. Pot1 deficiency impairs cell cycle progression and triggers the production of ECTR. A.** A *pot1^crg1* liquid culture grown in YPA was harvested, washed 3 times with water, and resuspend in YPD to a starting $OD_{600}$ of 0.1. The population doubling of the culture was then monitored over a period of 48 hours. **B.** The UCM350 and *pot1^crg1* strains were first grown in YPA, and then switched to YPD medium. After another 24 hours of growth, the cells were examined under the microscope. **C.** Chromosomal DNAs from the indicated strains grown in either YPA or YPD were isolated and subjected to Southern analysis with a telomere repeat probe (TR82) with or without prior *Pst*I digestion. **D.** DNAs from *pot1^crg1* and *pot1ΔN^crg1* grown in YPD were treated with the indicated nucleases and then subjected to in-gel hybridization analysis of G- and C-strand ssDNA. The signals were normalized against the untreated sample, and then plotted on the right. **E.** The *pot1* RNA levels from the indicated strains grown in the specified media were analyzed by RT-qPCR. The results were normalized against the signal for UCM grown in YPD, and then plotted.
(TIF)

**S6 Fig. Telomere-telomere fusions in *pot1*-deficient *U. maydis*. A.** (Top) Schematic diagram of the primers and probes used to detect fusions between UT4- and UT6-containing telomeres. (Bottom) Chromosomal DNAs from the indicated strains grown in either YPA or YPD were subjected to PCR-based fusion detection. Five independent PCR reactions per DNA sample were performed; the products were separated by electrophoresis and subjected to Southern analysis using sequentially a UT4 and a UT6 probe. **B.** The number of fusion fragments were determined and plotted. Statistical significance was calculated using Students' test (*, <0.05; **, <0.01; ***, <0.001).
(TIF)

**S7 Fig. The growth of *pot1*, *rad51*, and *brh2* single and double mutants. A.** Serial dilutions of the indicated strains were spotted on YPA and YPD plates. The growth of the strains after 2 and 3 days of incubation were imaged. **B.** The morphologies of the indicated strains after 24 hours of growth in YPD were examined.
(TIF)

**S8 Fig. Pot1 deficiency triggers the accumulation of telomere repeat containing RNAs.** (Left) Schematic illustrations of the structure of UT6 telomeres and the primers used in the RT and PCR reactions designed to detect telomere repeat RNAs. Note that ARRET was previously defined as C-strand RNA comprised of subtelomere sequences only. (Right) RNA samples from the indicated strains grown in YPA or YPD were subjected to RT-PCR using the primers indicated to the left of each panel. Both the F1/R2 and F2/R1 PCR reactions are expected to generate ~200 bp products. The multiple bands detected are likely due to minor insertions/

deletions in different copies of UT6 elements [22].
(TIF)

**S9 Fig. The nuclease sensitivity of RNA species from telomeres in pot1-deficient *U. maydis*.** (Top) RNAs from the indicated strains grown in YPD were treated with the indicated nucleases and then subjected to dot blot analyses using sequentially probes for detecting G-strand RNA, C-strand RNA, and 26S rRNA. (Bottom) The ratios of G-strand and C-strand signals to rRNA signals for the untreated, DNase-treated and RNase H-treated samples were calculated. Each ratio was normalized against the untreated sample for the particular strain and plotted. (TIF)

**S10 Fig. Possible evolutionary relationships between three OB-fold rich complexes at telomeres.** The domain structures of RPA, CST and PT subunits are illustrated schematically. The designations for the domains are as follows: OB, oligosaccharide/oligonucleotide-binding; WH, winged-helix; HRJ, Holliday junction resolvase. The OB folds implicated in ssDNA-binding are shaded in dark brown. Available structural and functional evidence strongly suggests that RPA and CST share a common ancestry. Whether PT is derived from the same primordial ssDNA-binding complex is less clear. The demonstration of a 4-OB architecture for Pot1 orthologs in fungi and metazoan suggests that Pot1 and RPA1 have similar domain organizations and may indeed be evolutionarily related. (TIF)

**S1 Table. *U. maydis* strains used in this study.** (DOCX)

**S2 Table. Oligos used in this study.** (DOCX)

**S3 Table. Pot1 homologs utilized for *in silico* analysis in this study.** (DOCX)

**S1 Data. Numerical data for all graphs.** (XLSX)

## Acknowledgments

We thank Qingwen Zhou for purified Rad51 and Brh2/Dss1, and members of our laboratories for discussion.

## Author Contributions

**Conceptualization:** William K. Holloman, Neal F. Lue.

**Data curation:** Syed Zahid, Sarah Aloe, Jeanette H. Sutherland, Neal F. Lue.

**Funding acquisition:** Neal F. Lue.

**Investigation:** Syed Zahid, Sarah Aloe, Jeanette H. Sutherland, Neal F. Lue.

**Methodology:** William K. Holloman.

**Project administration:** Neal F. Lue.

**Resources:** William K. Holloman.

**Supervision:** William K. Holloman, Neal F. Lue.

**Writing – original draft:** Neal F. Lue.

**Writing – review & editing:** Syed Zahid, William K. Holloman, Neal F. Lue.

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
