## [Decision Letter · Decision Letter 0]

13 Jan 2022

Dear Dr Lue,

Thank you very much for submitting your Research Article entitled 'Ustilago maydis telomere protein Pot1 harbors an extra N-terminal OB fold and regulates homology-directed DNA repair factors in a dichotomous and context-dependent manner' to PLOS Genetics.

The manuscript was fully evaluated at the editorial level and by independent peer reviewers. The reviewers appreciated the attention to an important problem, but raised some substantial concerns about the current manuscript. Based on the reviews, we will not be able to accept this version of the manuscript, but we would be willing to review a much-revised version. We cannot, of course, promise publication at that time.

If you decide to revise the manuscript for further consideration at PLOS Genetics, please aim to resubmit within the next 60 days, unless it will take extra time to address the concerns of the reviewers, in which case we would appreciate an expected resubmission date by email to plosgenetics@plos.org.

[LINK]

We are sorry that we cannot be more positive about your manuscript at this stage. Please do not hesitate to contact us if you have any concerns or questions.

Yours sincerely,

Jin-Qiu Zhou

Associate Editor

PLOS Genetics

Gregory P. Copenhaver

Editor-in-Chief

PLOS Genetics

Reviewer's Responses to Questions

**Comments to the Authors:**

Reviewer #1: This manuscript by Zahid et al. uses both biochemical and genetic approaches to examine the role of the telomeric protein Pot1 in the fungus U. maydis. The data provide an important step forward in understanding the role of this protein at telomeres in U. maydis, a model organism with arguably more telomeric similarities to metazoans than most other fungal model systems. Although most of the results reported bear some similarities to past results in mammalian or other systems, the manuscript should still be of quite broad interest in the telomere field. The manuscript is well written and the results presented are generally of high quality. Some specific criticisms are given below.

P2, l 47-48: This sentence should be altered a bit. I would not say that Rad51 is involved in completing telomere replication as this has not been shown directly here. Length regulation may be perturbed but this is not the same thing.

P3, l 67-70: I suggest modifying the description of telomere erosion. Replication fork blockages by themselves do not shorten telomeres. Presumably, the subsequent processing is doing the actual shortening. Also, substantial erosion will occur via truncations from damage or recombination that are likely independent of replication fork blocks.

It would be useful to mention in the text early on what the sequence and lengths of U. maydis telomeres are.

P5: Some additional information and perhaps analyses should be added regarding the discussion of the additional OB-fold (OB-N). One question is whether five predicted beta folds separated by an alpha helix is adequate as defining an OB-fold. The two predicted structures shown (Fig. 1C) seem rather different from one another and not obviously equivalent (to me at least) to many established beta barrel OB folds and the author’s own evidence indicates that OB-N isn’t involved in DNA binding. Also, where the predicted secondary structural features are should be clearly indicated in Fig. 1B (separately for each of the two species where a predicted structure is shown). Another matter is the question of the origin of the extra OB-fold. The authors favor the idea that it is an ancestral trait, conserved with the N-terminal OB-fold of metazoans. However, the apparent lack of this domain in each of other, more basal groups of fungi certainly disfavors this hypothesis. The authors should compare the sequence of OB-N to that of OB-1 and OB-2. One possibility is that OB-N in basiomycetes represents an independent acquisition of a fourth OB-fold either from an ancient duplication of one of the other OB-folds in Pot1 or perhaps even from some other protein. Comparison of OB-N to OB-1 and OB-2 could help address at least the former possibility. The authors should also discuss a bit more their search comparisons to other fungi. Have all available fungi in Ascomycota, Zoopagomycota and Chytridiomycota been examined? Finding even a single non-basiomycete with the OB-N could greatly strengthen the possibility of multiple independent loss events of OB-N in these other fungal groups. The authors might also want to mention the possibility that an independent acquisition of an OB-N in basiomycetes could conceivably be accompanied by some degree of convergent evolution that might lead to some resemblance to the metazoan OB-N.

P5-6: Some details of the strand-exchange assay should be mentioned in the text. Minimally, the authors should describe whether the assay shown involves telomeric or non-telomeric sequences and what the sequences of the oligos used was. I don’t know if there are technical problems with examining strand exchange with repetitive sequences but doing the reaction with telomeric and non-telomeric repeats would be very interesting. Oligos with either the ss or the ds region telomeric could be used. One might guess that Pot1 would be better at causing strand exchange with telomeric compared to non-telomeric sequences, especially since the reported effect in the data here is quite modest. Strand exchange that favored telomeric sequence would be both especially interesting and harder for skeptics to ignore.

Define the asterisks in the legend to Fig. 2D.

The authors should discuss the additional bands seen in the strand exchange assay, most notably in Fig. 2D and what evidence proves the particular band expected to be the result of strand exchange is actually that. An arrow pointing to this specific band would be helpful too. At least one of the other bands varies in intensity quite considerably, increasing when Pot1 is present.

P6: The variable telomere length phenotype of U. maydis wild type cells complicates trying to prove modest average length changes but the data presented seem adequate for arguing that the Pot1 mutation and the Rad51 mutation do indeed have somewhat shorter telomeres than wild type. The authors argue that the physical interactions of Pot1 and Rad51 and the apparently similar telomere shortening in pot1 delta N and rad51 mutants suggests that these proteins act in the same telomere maintenance pathway. However, the authors’ data in Fig. 5 suggests that brh2 delta mutants also may have similarly shortened telomeres. This ought to be further examined. If true, this would add more information to help elucidate the connection between HR and telomeres in U. maydis.

P7/Fig. 4B: It would be worthwhile to have quantification of the relative telomere signal in the mutants. While this would be subject to some issues regarding interpretation, it would still be interesting to estimate.

Fig. 5C: G circle and C circle data are stated as being shown but only C circle data is shown.

Fig. 6B legend appears to refer to Fig. 6D.

Reviewer #2: The review is uploaded as an attachment.

Reviewer #3: In this manuscript, Dr. Lue and his colleagues identified an extra N-terminal OB fold (OB-N) in fungal Ustilago maydis Pot1 (UmPot1). By sequence alignment, the authors showed that this OB-N is present in most metazoans. Using in vitro biochemical studies, the authors further demonstrated that UmPot1 OB-N, together with OB1 and OB2, binds directly to Rad51 and regulates its strand exchange activity. Deletion of OB-N domain alone resulted in telomere shortening, while deletion of Pot1 through transcriptional repression resulted in growth arrest and severe telomeric structural aberrations as well as high level of TERRA. In addition, the phenotypes can be largely suppressed by co-deletion of rad51 and brh2. Thus, the authors conclude that UmPot1 maintains telomere integrity under both normal and deprotected states through regulation of two major HDR repair proteins Rad51 and Brh2. Overall, the experiments were well designed and carefully characterized. The data is of high quality and presented in a clear and convincing manner. I have a few comments listed below:

1. The authors mentioned evolutionary relationships of RPA like complexes at telomeres. It would be interesting to check the sequence/2D/3D structural similarity between OB-N domain in UmPot1 and the OB fold in RPA1 or CTC1 to get an idea whether UmPot1 is a “chimeric” RPA-Pot1 form.

2. In Figure 3, TRF Southern blot showed Pot1∆N strain had clear telomere shortening, while the deletion seems not affecting overall growth by serial dilution. Have the authors followed a long-term growth effect of Pot1∆N strain? Or in the presence of replication stress regents like HU to see whether it accelerates telomere shortening and senescence?

3. The deletion of Pot1 (Pot1crg1 and Pot1∆Ncrg1) resulted in a substantial increase of ssDNA shown in Figure 4. Are those ssDNA generated by aberrant HDR or by resection, or are those ssDNA located internally or terminally? Is it possible to distinguish it by adding the Exo I digestion?

4. The OB-N is present in most basidiomycete, but not in fission yeast and other filamentous fungi. Have the authors considered generating a chimeric fusion Pot1 with the OB-N from UmPot1 fused with other fission yeast or filamentous fungi Pot1?

Minor points:

1. In Figure 1D, it will be easier for the readers to have the scheme of the truncation variants as shown in S2 figure a.

2. From the biochemical binding experiment, have the authors calculate the stoichiometry ratio of Pot1 and Rad51?

**Have all data underlying the figures and results presented in the manuscript been provided?**

Reviewer #1: Yes

Reviewer #2: Yes

Reviewer #3: Yes

PLOS authors have the option to publish the peer review history of their article (what does this mean?). If published, this will include your full peer review and any attached files.

Reviewer #1: No

Reviewer #2: No

Reviewer #3: No

---

## [Decision Letter · Decision Letter 1]

2 Apr 2022

Dear Dr Lue,

We are pleased to inform you that your manuscript entitled "Ustilago maydis telomere protein Pot1 harbors an extra N-terminal OB fold and regulates homology-directed DNA repair factors in a dichotomous and context-dependent manner" has been editorially accepted for publication in PLOS Genetics. Congratulations!

Yours sincerely,

Jin-Qiu Zhou

Associate Editor

PLOS Genetics

Gregory P. Copenhaver

Editor-in-Chief

PLOS Genetics

Comments from the reviewers (if applicable):

Reviewer's Responses to Questions

**Comments to the Authors:**

Reviewer #1: The authors have addressed all my criticisms and I have no further criticisms.

Reviewer #2: Zahid et al. has now revised the manuscript with additional data shown in Figures 1 - 3, S2, S4, and S5. The authors have also revised the text to address the reviewers critiques. I do not have any further concerns about this manuscript.

Reviewer #3: In this revised manuscript, the authors have addressed majority of my questions. I am particularly impressed by the new S5 Figure D that showed both resection and resolution of HDR involved in generation of ssDNA. The overall manuscript is much improved. Given the importance of the topic, it will appeal to a wide audience in the telomere field.

**Have all data underlying the figures and results presented in the manuscript been provided?**

Reviewer #1: Yes

Reviewer #2: Yes

Reviewer #3: Yes

PLOS authors have the option to publish the peer review history of their article (what does this mean?). If published, this will include your full peer review and any attached files.

Reviewer #1: No

Reviewer #2: No

Reviewer #3: No

**Data Deposition**

http://datadryad.org/submit?journalID=pgenetics&manu=PGENETICS-D-21-01605R1

**Press Queries**

---

## [Editor Report · Acceptance letter]

22 Apr 2022

PGENETICS-D-21-01605R1 

Ustilago maydis telomere protein Pot1 harbors an extra N-terminal OB fold and regulates homology-directed DNA repair factors in a dichotomous and context-dependent manner 

Dear Dr Lue, 

We are pleased to inform you that your manuscript entitled "Ustilago maydis telomere protein Pot1 harbors an extra N-terminal OB fold and regulates homology-directed DNA repair factors in a dichotomous and context-dependent manner" has been formally accepted for publication in PLOS Genetics! Your manuscript is now with our production department and you will be notified of the publication date in due course.

With kind regards,

Livia Horvath

PLOS Genetics

On behalf of:
